# Optimization of PVDF-TrFE Based Electro-Conductive Nanofibers: Morphology and In Vitro Response

**DOI:** 10.3390/ma16083106

**Published:** 2023-04-14

**Authors:** William Serrano-Garcia, Iriczalli Cruz-Maya, Anamaris Melendez-Zambrana, Idalia Ramos-Colon, Nicholas J. Pinto, Sylvia W. Thomas, Vincenzo Guarino

**Affiliations:** 1Advanced Materials Bio & Integration Research (AMBIR) Laboratory, Department of Electrical Engineering, University of South Florida, Tampa, FL 33620, USA; 2Institute of Polymers, Composites and Biomaterials, National Research Council of Italy, Mostra d’Oltremare, Pad.20, 80125 Naples, Italy; 3Department of Physics and Electronics, University of Puerto Rico at Humacao, Humacao 00791, Puerto Rico

**Keywords:** electroconductive, nanofibers, composites, electrospinning

## Abstract

In this study, morphology and in vitro response of electroconductive composite nanofibers were explored for biomedical use. The composite nanofibers were prepared by blending the piezoelectric polymer poly(vinylidene fluoride–trifluorethylene) (PVDF-TrFE) and electroconductive materials with different physical and chemical properties such as copper oxide (CuO), poly(3-hexylthiophene) (P3HT), copper phthalocyanine (CuPc), and methylene blue (MB) resulting in unique combinations of electrical conductivity, biocompatibility, and other desirable properties. Morphological investigation via SEM analysis has remarked some differences in fiber size as a function of the electroconductive phase used, with a reduction of fiber diameters for the composite fibers of 12.43% for CuO, 32.87% for CuPc, 36.46% for P3HT, and 63% for MB. This effect is related to the peculiar electroconductive behavior of fibers: measurements of electrical properties showed the highest ability to transport charges of methylene blue, in accordance with the lowest fibers diameters, while P3HT poorly conducts in air but improves charge transfer during the fiber formation. In vitro assays showed a tunable response of fibers in terms of viability, underlining a preferential interaction of fibroblast cells to P3HT-loaded fibers that can be considered the most suitable for use in biomedical applications. These results provide valuable information for future studies to be addressed at optimizing the properties of composite nanofibers for potential applications in bioengineering and bioelectronics.

## 1. Introduction

Electroconductive polymers have garnered significant attention in recent years due to their unique combination of electrical conductivity, mechanical, and chemical properties. This makes them suitable for a wide range of applications in fields such as electronics, energy storage, and sensing [1,2,3]. In this context, research efforts have focused on exploring the properties and potential of various electroconductive polymeric composites and developing new methods for their synthesis and characterization for biomedical use [4,5,6,7].

Despite these efforts, there are still many questions that need to be addressed, particularly with regard to the optimization of polymer properties for specific applications such as bioelectronics. Furthermore, the development of new electroconductive polymers and materials with improved performance is of ongoing importance. This is particularly true for applications in demanding environments, where high levels of performance and stability are required [8,9,10].

In the last years, electroactive materials such as copper oxide (CuO) [11,12], poly(3-hexylthiophene) (P3HT) [13,14], copper phthalocyanine (CuPc) [15], and methylene blue (MB) [16] have emerged as promising candidates for biological purposes including biosensors, tissue engineering, and drug delivery. CuO has been extensively investigated as a potential material for biosensors due to its excellent electrocatalytic activity and high stability in biological environments [17,18]. For instance, P3HT, a p-type semiconducting polymer, has been used in tissue engineering applications as a neural scaffold material due to its biocompatibility, good electrical conductivity, and mechanical properties [19]. CuPc has shown promise in drug delivery due to its ability to encapsulate drugs and selectively release them in response to specific stimuli, such as light or pH [20,21,22]. MB, a redox-active dye, has been widely used in electrochemical biosensors due to its ability to act as an electron mediator and its excellent electrochemical activity [23,24,25,26].

The unique properties of these electroactive materials have enabled their use in a wide range of bioelectronics and biological applications. These applications include biosensors for monitoring various biomolecules and metabolites, tissue engineering for the regeneration of damaged tissues, and drug delivery for targeted and controlled release of therapeutic agents. 

In this context, the use of these materials embedded into nanofibers has the potential to revolutionize the biotechnology field. It can provide a foundation for future research efforts aimed at optimizing the properties of electroconductive polymers with piezoelectric platforms and expanding their use in various fields. Herein, a comparative study on composite piezoelectric PVDF-TrFE fibers embedded with electroactive phases, i.e., namely, CuO, P3HT, CuPc, and MB, was proposed in order to investigate the correlations among electroconductive properties, morphology, and in vitro response.

## 2. Materials and Methods

Nanofibers were fabricated using PVDF-TrFE (Mw = 35,000) dissolved in tetrahydrofuran (THF) at a 13 wt% solution, as published elsewhere. [27] The active materials used to perform the composite nanofibers were copper oxide (CuO) nanoparticles of 25 nm (partially passivated, 99.8%, 25 nm; US Research Nanomaterials, Houston, TX, USA), poly(3-hexylthiophene) (P3HT) (regioregular, ca. Mn 54,000–75,000, electronic grade, 99.995% trace metals basis; Sigma-Aldrich, St. Louis, MO, USA), copper phthalocyanine (CuPc) (sublimed grade, dye content 99%, Mw = 576.07; Sigma-Aldrich, St. Louis, MO, USA), and methylene blue (MB) (Mw = 319.85; Sigma-Aldrich, St. Louis, MO, USA). Subsequent composite fibers were formed by adding a selected amount of the active component for every 1.5 g of 13 wt% PVDF-TrFE/THF. Experimental amounts are shown in Table 1. Solutions were electrospun using 1.5 mL via a 23 G syringe, under a high voltage of 17 kV, with a pump rate of 300 µL/h, and collected on a slow-rotation mandrel (300 rpm) 13 cm apart.

The morphology of the nanofibers was investigated by using scanning electron microscopy, SEM (JEOL JSM-IT-100). Briefly, a properly sized nanofiber sample was mounted on a sample holder by carbon double adhesive tape. Selected images were used for quantitative analyses. In detail, fiber diameters were measured from 100 nanofibers randomly using the SEM built-in measurement system. Meanwhile, energy dispersive spectroscopy (JEOL JSM-IT-100: EDS) analysis was used for a qualitative estimation of the elemental composition of the nanofibers.

Electrical properties of the composites were evaluated by drop casting the composite solutions over interdigitated gold electrodes to reveal the bulk electrical conductivity in air, at laboratory conditions. The electrical characterization was performed by three sweeps: from 0 V to 5 V, 5 V to −5 V, and −5 V to 0. This sweep shows possible charge traps and redox properties in the drop-casted films promoted by the characteristic properties of the conductive phase and the randomness of the electroactive molecules in the piezoelectric polymer.

For in vitro assays, L929 cell line (fibroblasts derived from mouse, Sigma-Aldrich, St. Louis, MO, USA) were used. The L929 cells were cultured in a 75 cm^2^ cell culture flask in Dulbecco’s Modified Eagle Medium (DMEM, Sigma-Aldrich, Milan, Italy) supplemented with 10% of fetal bovine serum (FBS, Sigma-Aldrich, St. Louis, MO, USA), antibiotic solution (streptomycin 100 μg/mL and penicillin 100 U/mL, Sigma-Aldrich, Milan, Italy), and 2 mM of L-glutamine (Sigma-Aldrich, Milan, Italy). The cells were incubated at 37 °C in a humidified atmosphere with 5% of CO_2_ and 95% air. Before the assays, samples were cut and placed into 96-well cell culture and sterilized with UV light for 30 min.

The cytotoxicity of fibers was evaluated by using the XTT assay protocol (XTT, Roche Diagnostics Deutschland GmbH, Mannheim, Germany, purchased by Sigma-Aldrich) at 24 and 72 h. The L929 cells were seeded at a density of 1 × 10^5^ cells per well onto PVDF, P3HT, MB, CuO, and CuPc fibers. The cell culture plate (TCP) was used as a control. At each scheduled time, the culture medium was replaced with 100 μL of fresh medium and 50 mL XTT solution was added, followed by 4 h of incubation. After the incubation time, the supernatant was measured using a plate reader (Wallac Victor 1420, PerkinElmer, Boston, MA, USA) at 450 nm of absorbance. The XTT assays were based on the reduction of tetrazolium salt XTT by mitochondrial enzymes into soluble formazan dye by living cells. In order to evaluate morphology and cell spreading, L929 were investigated by confocal laser scanning microscopy (LSM 510, Carl Zeiss, Berlin, Germany). Cell Tracker™ Green CMFDA(Invitrogen, Life Technologies Corporation, Eugene, OR, USA) in phenol red-free medium at standard conditions. Then, samples were washed with PBS (phosphate-buffered saline) solution and incubated for 1 h in complete medium. After the incubation, samples were washed three times and cells were fixed with 4% of PFA. Before imaging, samples were stained with 1 μg/mL DAPI (4′,6-diamidino-2-phenylindole) for 5 min, and then washed. For proliferation tests, 6 × 10^3^ cells per well were seeded onto P3HT, MB, and CuPc electrospun fibers to then perform cell proliferation at 1, 3, and 7 days by using XTT assay. The resulting amount of formazan dye was directly proportional to the number of viable cells for each electroactive culture.

## 3. Results 

The combination of electroactive components with piezoelectric matrices represents an interesting strategy to fabricate innovative devices. These devices have customizable electroconductive properties as a function of the peculiar properties of the dispersing phases. In this context, electrospinning offers the opportunity to emphasize this effect by creating electrostatic interactions among polar groups at the surface interface.

Figure 1 shows PVDF-based fibers collected onto the aluminum foil for ca. 10 min. Qualitatively, different colors of composite were recognized as a function of the active phase embedded; by comparing SEM images of deposited fibers, it is possible to recognize a homogeneous spatial distribution of fibers onto the collector that are characterized by a slight preferential alignment, in all the cases, qualitatively ascribable to the interaction of PVDF polar groups with applied electric forces. A limited number of beads—in the case of CuO, P3HT, and CuPc—until to none—in the case of PVDF-TrFE and MB—was recognized, thus confirming the optimization of the process conditions shown in Figure 2. EDS analysis was used to identify the elemental composition of the nanofibers with each peak corresponding to the elements present in the composite. For CuO, it shows peaks corresponding to copper and oxygen, while CuPc shows corresponding peaks to copper and carbon with smaller amounts on nitrogen and oxygen. For P3HT and MB, peaks corresponding to the elements carbon and hydrogen were present, while sulfur and nitrogen were not detected. This might be due to the detector not being sensitive enough to detect the low concentrations of those elements in the collector. The aluminum peak at 1.5 keV came from the foil collector and could also mask the small signal from those elements at such low concentrations. A decrease in diameter was observed in the case of fibers embedded with electroconductive phases. In particular, PVDF-TrFE fibers showed an average fiber diameter of ca. 3.75 µm, while those of the composite fibers decreased by 12.43% for CuO (~3.28 µm), 32.87% for Cu Phthalocyanine (~2.52 µm), 36.46% for P3HT (~2.38), and a remarkable 63% for MB (~1.39 µm), as shown in Figure 3. Notably, the deposition of fibers without the use of rotating collectors led to a reduced evaporation rate, thus promoting a longer electrostatically stretch before landing over the collector, and, therefore, a smaller diameter at the sub-micrometric scale. In the case of composite fibers, the presence of electroactive components concurs to amplify the effect of electrostatic forces on fiber diameters, thus favoring the deposition of thinner fibers, with respect to pure ones.

Current–voltage (I–V) curves shown in Figure 4 were recorded on pure and composite PVDF-TrFE nanofibers embedded with different active components. This result is strictly related to the intrinsic electrical properties of different active components used and their interaction with the PVDF-TrFE matrix. In particular, the I–V measurements showed that PVDF-TrFE composites of MB and CuPc had moderate conductivity, with MB exhibiting a redox electron transfer mechanism and CuPc with a charge carrier hopping mechanism. Composites with CuO and P3HT had poor conductivity in air, with an electron and hole hopping mechanism, respectively.

Methylene blue shows the highest ability to transport charges, following CuPc, in accordance with fibers diameters being the thinnest for MB, and followed similarly by P3HT and CuPc. While P3HT poorly conducts in air, it improves charge transfer during the fiber formation, enhancing the formation and the decrease in fiber diameter [27,28].

The electrical conductivity of electroactive materials and their composites is determined by the movement of charge carriers through the material. The conductivity is affected by factors such as the material’s crystal and molecular structure, doping level, percolation, and presence of impurities or defects. Copper oxide exhibits semiconducting properties, and its conductivity mechanism involves the movement of electrons through the material. Poly(3-hexylthiophene) is a conjugated polymer with semiconductive properties, and its conductivity mechanism involves the movement of holes through the polymer chains. Copper phthalocyanine is a small molecule organic semiconductor, and its conductivity mechanism involves the movement of ambipolar charge carriers through the material. Methylene blue acts as a redox-active dye, and its conductivity mechanism involves the transfer of electrons between the dye molecules and the electrodes via a hopping mechanism. The conductivity mechanisms of these materials can be enhanced by increasing their degree of crystallinity, improving the alignment and packing of the molecules, and facilitating charge carrier hopping. Additionally, the conductivity of the studied composites is influenced by the percolation of the active material into the composite and the properties of the PVDF-TrFE.

When PVDF-TrFE is added to these materials, it can have various effects on their conductivity mechanisms, depending on the nature of the interaction between them, as a function of the specific properties of active components used. In particular, the addition of PVDF-TrFE acts as a binder that mechanically facilitates the formation of the nanofibers and a well-connected network of particles or molecules in the composite, which can improve charge transport through the material. Additionally, PVDF-TrFE can also act as a dopant, introducing additional charge carriers into the material and increasing its conductivity. The presence of PVDF-TrFE can also modify the surface chemistry of the electrodes, leading to changes in the energy-level alignment at the interface and affecting the electron injection and charge transfer processes.

In the case of composite PVDF-TrFE fibers, the electrical conductivity can be influenced by the morphology and microstructure of the composite material. The presence of PVDF-TrFE can promote the formation of a more ordered crystalline structure, which can improve charge transport and enhance the overall conductivity of the composite material [29]. Additionally, the piezoelectric properties of PVDF-TrFE can enhanced the stability of the embedded phases by reducing the solubility of the individual components and preventing leaching from the fiber body, in agreement with previous experimental evidence [30].

Taking into consideration the different response of composite PVDF-TrFE nanofibers in terms of morphology and electrical properties, in vitro studies were performed in order to validate their biocompatibility and suggest the best formulation for potential biomedical uses.

In vitro tests were performed on PVDF-TrFE and their composites embedded with CuO, P3HT, CuPc, and MB fibers firstly to investigate cell adhesion at 24 and 72 h (Figure 5). The results are presented with respect to the control (TCP) which represents the 100% of cell viability. After 24 h, the results showed significant differences when all the samples were compared with the TCP; however, more than 60%, in all cases, of cells were viable. However, after 72 h, CuO showed a cytotoxic effect on the L929 cells, with significant difference with respect to fibers without active materials (PVDF-TrFE). Therefore, this group was excluded for the next assay.

In vitro response was also confirmed by the morphological investigation performed via confocal microscopy (Figure 6a–c). The images—showing stained cells: nuclei (blue) and bodies (green)—confirmed the ability of cells to adhere onto the PVDF TrFE fibers after 24 h, with a comparable density and similar cell morphology independently in the presence of P3HT, MB, and CuPc. However, a higher presence of cell nuclei (blue) related to the presence of live cells was recognized in the case of P3HT-loaded samples (Figure 6a), followed by MB and CuPc (Figure 6b,c), respectively.

The trend was also confirmed by in vitro tests performed to evaluate cell proliferation (Figure 7). In this case, L929 cells were maintained in culture for 1, 3, and 7 days. P3HT and MB fibers showed good cell proliferation with respect to CuPc fibers. Moreover, there was an increase in cell proliferation onto P3HT and CuPc fibers after 3 days; however, there was a stop in cell proliferation after 7 days. Meanwhile, the proliferation of MB fibers remained similar from the first to the third day and increased after 7 days.

In other terms, cell viability and proliferation assays showed that L929 cultured on P3HT and MB composites had higher viability and proliferation rates compared with those cultured on CuO and CuPc composites. These results are consistent with previous studies that have shown that P3HT and MB can promote cell adhesion and proliferation due to their surface properties and charge transfer mechanisms [31,32]. In contrast, CuO and CuPc, partially, have been shown to exhibit cytotoxicity and negative effects on cell viability due to their oxidative stress and reactive oxygen species generation [33]. The addition of PVDF-TrFE to each composite did not significantly affect cell viability or proliferation rates.

These promising results demonstrated a good biocompatibility of composite meshes, able to successfully support adhesion and proliferation of fibroblasts. Accordingly, further investigations could be performed in order to evaluate also the antibacterial response for a potential application of PVDF-TrFE composite nanofibers for physical filtration and/or static entrapment of bacteria.

## 4. Discussion

PVDF-TrFE electrospun nanofibers are recently emerging as interesting substrates with piezo/electroconductive properties suitable for a wide range of applications in the biomedical area. Indeed, the use of PVDF-TrFE provides the strength, flexibility, and mechanical stability needed to optimize the morphology of nanofibers for scaffolds and/or bio-functional platforms. The piezoelectric and triboelectric properties of PVDF allow for the generation of electrostatic charges, which can be used to trap particles [34,35], molecular species [36], or microorganisms enhanced with the addition of electroactive phases (i.e., virus, bacteria [37,38,39,40,41,42,43]). In this work, the optimization of composite nanofibers by embedding active components with electroconductive properties into PVDF-TrFE nanofibers was investigated in order to investigate the effect of different conductive properties, related to different active components, on fiber morphology and in vitro response. In vitro results suggest that the conductive mechanisms ascribable to peculiar interactions among PVDF TrFE nanofibers and selected active components such as P3HT or MB play an important role in determining fiber morphological properties and in vitro response of L929 cells, in terms of adhesion and cell proliferation, in contrast with other compounds (i.e., CuO, CuPc) that show a partial cytotoxic effect. This result is not surprising but in agreement with previous experimental evidence, indicating the preferential use of CuO or CuPc for diagnostic and/or therapeutic use for cancer applications [44,45]. At this preliminary stage of the study, P3HT-loaded fibers can be considered the most suitable for use in biomedical applications, but further studies will be needed to fully elucidate the mechanisms underlying these effects, in order to finely optimize the design of fibers for specific applicative targets.

## 5. Conclusions

In conclusion, the present study explored the morphology and in vitro response of electroconductive composite nanofibers for biomedical applications. Blending PVDF-TrFE with different electroconductive materials such as CuO, P3HT, CuPc, and MB resulted in unique combinations of electrical conductivity, biocompatibility, and other desirable properties. Morphological analysis revealed that the fiber size was affected by the type of electroconductive phase used. The in vitro assays demonstrated a tunable response of the fibers in terms of cell viability and proliferation, with P3HT-loaded fibers showing the most suitable interaction with fibroblast cells. These findings provide valuable information for researchers and engineers in the field of biomedical engineering and electronics, which can help optimize the properties of composite nanofibers and expand their use in bioelectronics. Additionally, further investigations could be performed to evaluate the antibacterial response of PVDF-TrFE composite nanofibers for physical filtration and/or the static entrapment of bacteria. Overall, the study highlights the potential of electroconductive composite nanofibers as a promising material for a broad application in the biomedical area, from tissue engineering to biosensors, to biofiltration.

## Figures and Tables

**Figure 1 materials-16-03106-f001:**
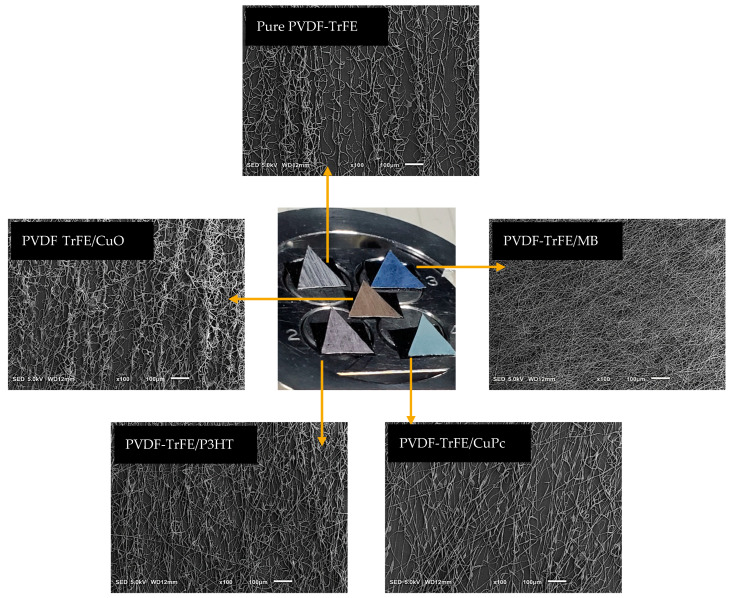
Different PVDF-TrFE electrospun fibrous membranes with electroconductive phases: qualitative evaluations.

**Figure 2 materials-16-03106-f002:**
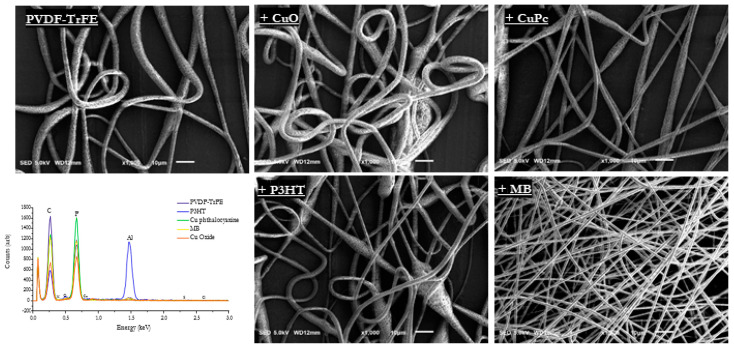
Comparison of pure and composite PVDF-TrFE nanofibers: SEM and EDS analysis.

**Figure 3 materials-16-03106-f003:**
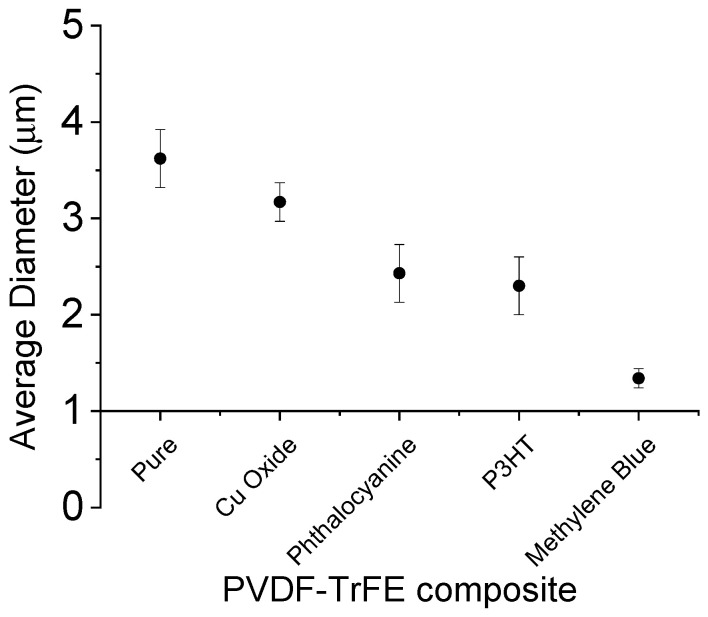
Quantitative evaluation of pure and composite PVDF-TrFE nanofibers’ fiber diameters via image analysis.

**Figure 4 materials-16-03106-f004:**
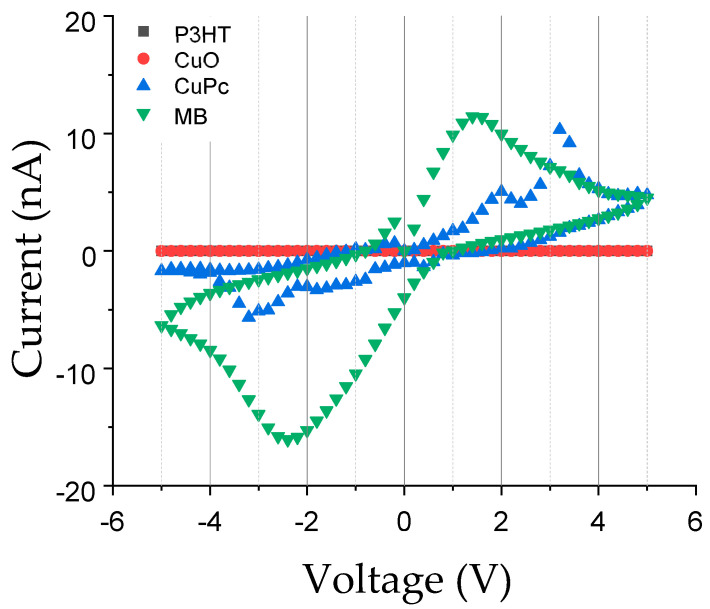
Electrical characterization of pure and composite PVDF-TrFE nanofibers: Current–voltage (I–V) curves as a function of different active components used.

**Figure 5 materials-16-03106-f005:**
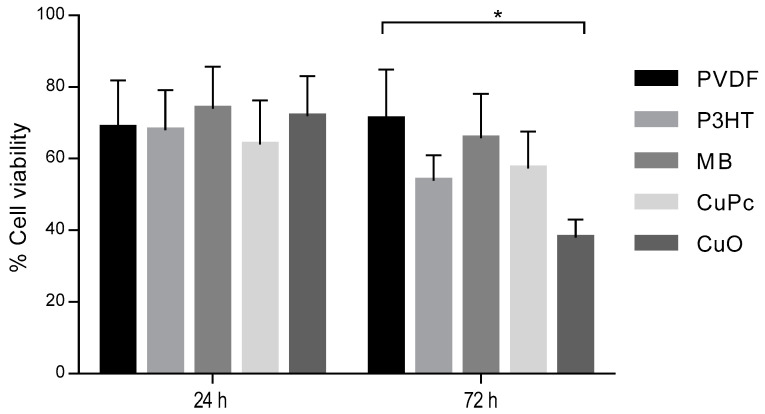
Comparative in vitro studies of PVDF, P3HT, MB, CuPc, and CuO electrospun fibers: cytotoxicity tests. All the results were normalized with respect to the positive control (TCP) representing the 100% of cell viability. (* *p* < 0.05).

**Figure 6 materials-16-03106-f006:**

Confocal microscopy of L929 cells’ adhesion after 24 h, onto electrospun fibers of PVDF-TrFE loaded with P3HT (**a**), MB (**b**), and CuPc (**c**). The images (Scale bar: 250 μm) were obtained by staining cells via Cell-Tracker Green CMFDA.

**Figure 7 materials-16-03106-f007:**
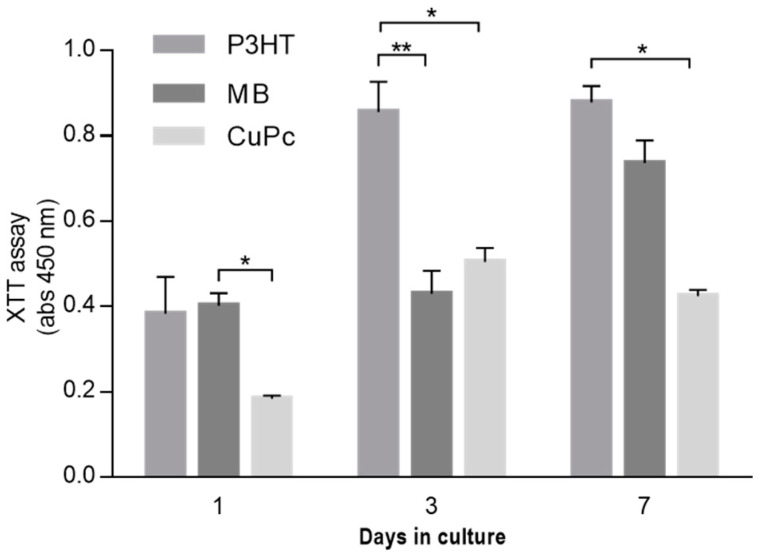
Cell proliferation of L929 cells seeded onto PVDF composites of P3HT, MB, and CuPc electrospun fibers. (* *p* < 0.05; ** *p* < 0.01). All the results were normalized with respect to PVDF fibers used as positive control.

**Table 1 materials-16-03106-t001:** Specifications of composite solutions.

Active Component	Weight (mg)	Amount of 13 wt% PVDF-TrFE/THF (g)	Composite Solution (wt%)
P3HT ^1^	1.4	1.45	0.09
Cu Phthalocyanine	10.9	1.5	0.721
Methylene Blue	10.2	1.5	0.675
CuO	8.9	1.5	0.589

^1^ 0.14 g of 1 wt% P3HT/THF was added to 1.3 g of 13 wt% PVDF-TrFE/THF.

## Data Availability

Not applicable.

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
