# Peer review of "Optimization of PVDF-TrFE Based Electro-Conductive Nanofibers: Morphology and In Vitro Response"

_materials, 2023, doi:10.3390/ma16083106_

Round 1
Reviewer 1 Report
The paper by Serrano-Garcia et al. proposed a comparative study on composite PVDF-TrFE fibers embedded with electroactive phases. Some problems should be addressed before its acceptance.
1. Please be sure that your abstract and your conclusions section not only summarize the key findings of your work but also concisely state the specific ways in which this work fundamentally advances the field relative to prior literature.
2. Actually, there're lots of publications on the electroconductive polymer composites, e.g., Polymer Engineering & Science, 2019, 59(S2), E224-E230; Polymers 2019, 11(3), 546. So appropriate citation and discussion should be added in the Introduction part.
3. The English and grammars should be well polished.
4. Authors should discuss why choose CuO, CuPc, P3HT, MB as conductive filler? More importantly, authors should discuss the novelty of this work in more detail in the introduction part.
5. Author should provide more information about the materials used in this study, such as the purity of CuO and MB, the molecular weight of PVDF-TrFE and P3HT, stereoregularity of P3HT and so on.
6. Author should provide the data of conductivity of electrospun fibrous membranes, and discuss why the diameter of composite fibers in the order: CuO > Cu Phthalocyanine > P3HT> MB.
7. The EDS image is too vague.
8. Line 137-140. “For P3HT and MB, peaks cor-137 responding to the elements carbon and hydrogen were present while sulfur and nitrogen 138 were not detected. This might be due to the detector no being sensitive enough to detect 139 the low concentrations of those elements in the collector”, This explanation is not convincing, the author should provide XPS data to confirm the presence of S , N elements.
9. Line 187-206, this part of content is common sense, and there is no need to elaborate.
10. Line 231, what is the control ? author should clearly express this concept.
Author Response
The paper by Serrano-Garcia et al. proposed a comparative study on composite PVDF-TrFE fibers embedded with electroactive phases. Some problems should be addressed before its acceptance.
Thank you so much for your time and comments on our work.
- Please be sure that your abstract and your conclusions section not only summarize the key findings of your work but also concisely state the specific ways in which this work fundamentally advances the field relative to prior literature.
Thank you for the comment. The last sentence of the abstract has been amended in order to better emphasise the future perspectives of the proposed study.
- Actually, there're lots of publications on the electroconductive polymer composites, e.g., Polymer Engineering & Science, 2019, 59(S2), E224-E230; Polymers 2019, 11(3), 546. So appropriate citation and discussion should be added in the Introduction part.
Thank you for the suggestion. “Wang, Y.; Yu, H.; Li, Y.; Wang, T.; Xu, T.; Chen, J.; Fan, Z.; Wang, Y.; Wang, B. Facile Preparation of Highly Conductive Poly(amide-imide) Composite Films beyond 1000 S m−1 through Ternary Blend Strategy. Polymers 2019, 11, 546. https://doi.org/10.3390/polym11030546 has been included as ref 8.
- The English and grammars should be well polished.
Thank you so much. The manuscript has been edited accordingly.
- Authors should discuss why choose CuO, CuPc, P3HT, MB as conductive filler? More importantly, authors should discuss the novelty of this work in more detail in the introduction part.
Thank you for the observation. We have edited the introduction part accordingly by adding our contribution on the usage of piezoelectric-electroconductive composite towards in-vitro characterizations.
- Author should provide more information about the materials used in this study, such as the purity of CuO and MB, the molecular weight of PVDF-TrFE and P3HT, stereoregularity of P3HT and so on.
Thank you for the observation. We have included the information as requested.
- Author should provide the data of conductivity of electrospun fibrous membranes, and discuss why the diameter of composite fibers in the order: CuO > Cu Phthalocyanine > P3HT> MB.
Thank you for the observation. We appreciate your suggestion regarding the conductivity data of our electrospun fibrous membranes and the discussion of the relationship between the diameter of composite fibers and the electroactive phases. However, we would like to clarify that we did not perform conductivity measurements for our fibrous membranes in this study and only evaluated the drop casted solutions for the composites in air. Furthermore, in order to characterize different diameters will require the preparation of other solutions to see the gradual effect of the amount of the electroconductive material vs. fiber diameter.
Although we did not investigate the relationship between the diameter of composite fibers and the electroactive phases in this study, we are currently designing future experiments where we investigate the behavior of the electroactive concentration vs. fiber diameter and the percolation conductivities. We will consider addressing this question in future studies and thank you for bringing it to our attention.
We have made the necessary revisions to the manuscript to better clarify the limitations of our study and the areas for future research. We hope that these changes are satisfactory, and we appreciate your feedback.
- The EDS image is too vague.
Thank you for this comment. More details about morphological analyses and EDS were included into the manuscript. Moreover, comments on EDS images have been improved.
- Line 137-140. “For P3HT and MB, peaks cor-137 responding to the elements carbon and hydrogen were present while sulfur and nitrogen 138 were not detected. This might be due to the detector no being sensitive enough to detect 139 the low concentrations of those elements in the collector”. This explanation is not convincing, the author should provide XPS data to confirm the presence of S , N elements.
Thank you for your comment, XPS analysis is not available in our labs. However, we tried to improve the discussion into the manuscript in order to clarify the EDS data.
- Line 187-206, this part of content is common sense, and there is no need to elaborate.
Thank you for the comment. This part of the text has been summarized in order to leave some information useful for the discussion.
- Line 231, what is the control? author should clearly express this concept.
Thank you for the comment, the control has been now specified into the main text and also reported in the caption of figure 5.
Control for the in-vitro test (and/or in figure 7)?
PVDF fibers have been taken as positive control, the results have been normalized respect to them.This has been reported in the caption of figure 7
Reviewer 2 Report
This paper presented aspects regarding the preparation of electroconductive composite nanofibers based on polymer poly(vinylidene fluoride–trifluorethylene) (PVDF-TrFE) and the study of the morphology and in vitro response of these composite nanofibers. The composite nanofibers were obtained by electrospinning of different blends of the PVDF-TrFE and electroconductive materials such as copper oxide, poly(3-16 hexylthiophene), copper phthalocyanine, and methylene blue. The results provide information about the optimization of the composite nanofibers properties and the possibility of being used in bioelectronics.
The paper is sound, original, interesting, well-organized and clearly presented.
The references are current and in accordance with the subject approached.
The following references can also be taken into account:
https://doi.org/10.1515/ntrev-2022-0082
https://doi.org/10.3390/polym14204311
Be careful with the numbering and formatting of the references.
I consider that the paper is of interest, suitable for publication in Materials and it can be considered for publication.
Author Response
This paper presented aspects regarding the preparation of electroconductive composite nanofibers based on polymer poly(vinylidene fluoride–trifluorethylene) (PVDF-TrFE) and the study of the morphology and in vitro response of these composite nanofibers. The composite nanofibers were obtained by electrospinning of different blends of the PVDF-TrFE and electroconductive materials such as copper oxide, poly(3-16 hexylthiophene), copper phthalocyanine, and methylene blue. The results provide information about the optimization of the composite nanofibers properties and the possibility of being used in bioelectronics.
The paper is sound, original, interesting, well-organized and clearly presented.
Thank you so much for your time and comments on our work.
The references are current and in accordance with the subject approached.
Thank you so much. We highly appreciate it.
The following references can also be taken into account:
https://doi.org/10.1515/ntrev-2022-0082
https://doi.org/10.3390/polym14204311
Thank you for the comments, references have been included as refs 9 and 10
Be careful with the numbering and formatting of the references.
Checked
I consider that the paper is of interest, suitable for publication in Materials and it can be considered for publication.
Thank you for your time and consideration.
Reviewer 3 Report
Review “Optimization of PVDF-TrFE based electro-conductive Nanofibers: Morphology and in vitro response”
Fibers were produced by electrospinning from different solutions based on PVDF-TrFE and four additives CuO, CuPc, P3HT and MB. The influence of additives on film conductivity, fiber morphology, fiber diameter, toxicity and cell adhesion and proliferation was determined. MB composite had the finest fibers. All but CuO after 72h passed a cell viability measure around 60%.MB and P3HT composites had good cell adhesion and proliferation. Knowledge of influence of electro-conductive additives on nanofiber cell interactions are important for device development and other applications in the biomedical field.
The research point is of interest for the biomedical field. The article is well written.
Some comments:
· Materials and methods: information about image analysis for diameter determination is missing. How many images were analyzed? Were images from different samples analyzed? Beads were ignored?
· Line 128 very long sentence which should be split. Check the whole paper for too long sentences.
· Abbreviation EDX was used in line 134 and EDS in Fig. 2 caption.
· Line 133 The term “used Figure 2” is not fitting the sentence.
· Table 1: all units have brackets “()” but not wt%
· Lines 147-151: These results are not shown in Fig. 3
· Fig. 2 Quality of EDX analysis graph is not good enough. Labels and curves are not visible, especially not in black/white printout. A stacked graph is suggested.
· Fig. 4 could have Fig. 4b with curves P3HT and CuO magnified. They overlap in Fig 4 and they are almost zero.
· Fig. 5 caption, Typo in the word toxicity
· Fig. 5 and 6, could the numbers on the x-axis not be rotated 45 degrees as it makes it difficult to read these numbers
· References: Why do all references have a number at the beginning like 1, W1, W2, 2 for ref. 1-4. These numbers should be removed. The references should not be copied from another publication.
· Maybe for future. Could the findings be justified with models or simulations? Could there be a graph illustrating the connections between electro-conductive properties and fiber diameter, cell viability etc.

Author Response
Fibers were produced by electrospinning from different solutions based on PVDF-TrFE and four additives CuO, CuPc, P3HT and MB. The influence of additives on film conductivity, fiber morphology, fiber diameter, toxicity and cell adhesion and proliferation was determined. MB composite had the finest fibers. All but CuO after 72h passed a cell viability measure around 60%.MB and P3HT composites had good cell adhesion and proliferation. Knowledge of influence of electro-conductive additives on nanofiber cell interactions are important for device development and other applications in the biomedical field.
The research point is of interest for the biomedical field. The article is well written.
Thank you so much for your time and comments on our work.
Some comments:
- Materials and methods: information about image analysis for diameter determination is missing. How many images were analyzed? Were images from different samples analyzed? Beads were ignored?
Thank you for your observation. We have added a sentence on how we measured the average diameters of the nanofibers. The images for each electroconductive material were analysed by measuring the diameters of the nanofibers directly on the SEM while at a magnification of x1000. Since the nanofibers were electrospun using a concentration beads were not formed, the elongation in the fibers that might reassemble elongated beads are taken into account and part of the error-bar in Figure 3.
- Line 128 very long sentence which should be split. Check the whole paper for too long sentences.
Thank you for your observation. We have edited the document to rectify too long sentences while trying to remain as clear as possible.
- Abbreviation EDX was used in line 134 and EDS in Fig. 2 caption.
Thank you, the text has been uniformed
- Line 133 The term “used Figure 2” is not fitting the sentence.
Thank you so much for the observation. We have changed “used” for “shown in”.
- Table 1: all units have brackets “()” but not wt%
Thank you for the observation. We have added (wt%) to table 1.
- Lines 147-151: These results are not shown in Fig. 3
Thank you for your observation. We have included the approx. diameters in the sentence.
- Fig. 2 Quality of EDX analysis graph is not good enough. Labels and curves are not visible, especially not in black/white printout. A stacked graph is suggested.
Thank you for the comment. The figure was improved as requested
- Fig. 4 could have Fig. 4b with curves P3HT and CuO magnified. They overlap in Fig 4 and they are almost zero.
Thank you so much for your observation. The electrical characterizations were performed in air conditions and measured conductivities for P3HT and CuO were low enough for the multimeter to measure noise-type signal. We are currently designing future experiments were the percolation into the nanofibers is taken into account and to include measurements comparisons between air and vacuum.
- Fig. 5 caption, Typo in the word toxicity
Thank you. The word has been edited to “cytotoxicity”.
- Fig. 5 and 6, could the numbers on the x-axis not be rotated 45 degrees as it makes it difficult to read these numbers
Thank you for the comment, figures have been revised according to the suggestion.
- References: Why do all references have a number at the beginning like 1, W1, W2, 2 for ref. 1-4. These numbers should be removed. The references should not be copied from another publication.
Thank you for the observation. We have each reference attached to an intrinsic value like 1, W1, W2, 2, etc and to a variable value such as the one from the “Numbering list format” in Word to help us track the refences during edition and revision. We have removed the “intrinsic values” from the references, and these has been revised accordingly.
- Maybe for future. Could the findings be justified with models or simulations? Could there be a graph illustrating the connections between electro-conductive properties and fiber diameter, cell viability etc.
Thank you for this comment. In the manuscript, it has been provided a preliminary study to investigate the effect of conductive phases on biocompatibility to validate the use of electroconductive fibres for biomedical applications. Several studies suggest a correlation among fibre diameter and cell response (i.e., adhesion, proliferation) but a clear trend is really not defined, mainly due to the chemical composition of fibres that, case by case, can play a different role. In this perspective, further studies should be performed to evaluate more in details the contribution of electroconductive phases on cell morphology in order to more deeply explore the mechanism of cell interaction. This will allow to more properly model the correlations among conductive properties, fibre morphology and cell response.
Reviewer 4 Report
1. The originality of the study must be stated.
2. Line 75: Full form of THF should be given.
3. Was there any particular reason of using XTT assay instaed of MTT assay?
4. In vitro should be written in Italic.
Author Response
- The originality of the study must be stated.
Thank you for your comment. In this work, we preliminary study the potential use of PVDF/P3HT electrospun nanofibers as potential substrate with electroconductive properties for biomedical use. Similar nanofibers have been used for other applications such as triboelectric nanogenerators [Scientific Reports 2022, 12(1):14842], biofiltration [Materials Letters, 266, 2020, 127458], bioelectronics [Polymers 2022, 14(23), 5073; https://doi.org/10.3390/polym14235073]. To validate the in vitro response of nanofibers, we proposed to compare PVDF-TrFE nanofibers combined with other electroconductive materials such as CuO, CuPc, and MB. This was variously stated into the manuscript, in the introduction and conclusion sections.
- 2. Line 75: Full form of THF should be given.
Thank you, we have added the full form of THF as Tetrahydrofuran.
- Was there any particular reason of using XTT assay instaed of MTT assay?
Thank you for your question. XTT assay is generally more sensitive than MTT assay and can detect smaller changes in cell viability or cytotoxicity, which was of particular importance on our characterizations. Indeed, L929 cell line are known to produce high levels of endogenous reducing agents, which can interfere with MTT assay and result in false-positive or false negative results. XTT assay is less prone to interfere by endogenous reducing agents, as it used an electron-coupling reagent to measure cell viability.
- In vitroshould be written in Italic.
Thank you for the observation. We have changed it accordingly.
Round 2
Reviewer 1 Report
Authors answered all the questions.
Reviewer 4 Report
I have no more suggestions.